# Biocompatibility of Surface-Modified Membranes for Chronic Hemodialysis Therapy

**DOI:** 10.3390/biomedicines10040844

**Published:** 2022-04-03

**Authors:** Mario Bonomini, Luca Piscitani, Lorenzo Di Liberato, Vittorio Sirolli

**Affiliations:** 1Nephrology and Dialysis Unit, Department of Medicine, G. d’Annunzio University, Chieti-Pescara, SS. Annunziata Hospital, Via dei Vestini, 66013 Chieti, Italy; lorenzo.diliberato@asl2abruzzo.it (L.D.L.); vsirolli@unich.it (V.S.); 2Nephrology and Dialysis Unit, Department of Medicine, San Salvatore Hospital, Via Vetoio, 67100 L’Aquila, Italy; lucpis90@virgilio.it

**Keywords:** biocompatibility, hemodialysis, membrane, biomaterial, protein adsorption, coagulation, platelet, surface modification

## Abstract

Hemodialysis is a life-sustaining therapy for millions of people worldwide. However, despite considerable technical and scientific improvements, results are still not fully satisfactory in terms of morbidity and mortality. The membrane contained in the hemodialyzer is undoubtedly the main determinant of the success and quality of hemodialysis therapy. Membrane properties influence solute removal and the interactions with blood components that define the membrane’s biocompatibility. Bioincompatibility is considered a potential contributor to several uremic complications. Thus, the development of more biocompatible polymers used as hemodialyzer membrane is of utmost importance for improving results and clinical patient outcomes. Many different surface-modified membranes for hemodialysis have been manufactured over recent years by varying approaches in the attempt to minimize blood incompatibility. Their main characteristics and clinical results in hemodialysis patients were reviewed in the present article.

## 1. Introduction

End-stage renal disease (ESRD) requiring chronic renal replacement treatment represents a growing health problem all over the world. Natural replacement of renal function by kidney transplant is the best treatment option but is limited by the shortage of donor organs. Thus, most ESRD patients are treated via dialysis, i.e., hemodialysis (HD) or peritoneal dialysis, targeting patient survival and quality of life. Dialysis is a life-sustaining therapy for approximately 3.4 million people worldwide [1], a number which is projected to reach close to 5 million by 2025 [2]. Among dialysis therapies, HD is by far the most used modality.

HD is a renal replacement technique that uses a semipermeable polymeric membrane in an extracorporeal system to filter blood. Artificial membrane packed in the hemodialyzer represents the centerpiece of the entire HD therapy. It allows removal by diffusion and convection of retained low molecular weight (MW) toxic substances and excess water—which accumulate in blood because of kidney failure—providing life support for patients to live with variable degrees of rehabilitation. Removal of high MW substances may also occur by adsorption on the surface of certain membranes. In addition, blood–membrane interactions occurring during the HD procedure, which can lead to activation of various biochemical pathways and cells and which characterize the membrane’s bio(in)compatibility, have had increasingly focused clinical interest [3,4,5]. Bioincompatibility in HD refers to all the harmful effects elicited by blood contact with the membrane surface [6]. In the HD field, improving biocompatibility is a major challenge as we seek to improve the clinical outcome of ESRD patients [7,8].

The material of the dialysis membrane is the primary determinant of intradialytic biological reactions. Several new membrane materials aiming to improve the interactions with blood components have been developed over the years. This article reviewed the most recent technological advancements to improve the biocompatibility profile of HD membranes, focusing on results obtained by clinical use in vivo. Studies on materials under experimental development were not covered here and were reported in a recent extensive review [9]. The aim of the present article was to examine the biocompatibility performance of the new surface-modified membranes and their potential beneficial effects, as compared to conventional membranes, in patients on chronic HD therapy.

## 2. Biological Responses Triggered by Blood Contact with Membrane Material during HD

### 2.1. Adsorption of Plasma Proteins onto the Membrane Surface

The exposure of blood to a foreign surface in a medical device such as the membrane packed in the hemodialyzer results in almost instantaneous adsorption of plasma proteins [10]. This inevitable initial event governs all the subsequent biological reactions occurring during the HD procedure [8], and hence the biocompatibility of a membrane material. Adsorption of plasma proteins onto the surface of HD membranes is a complex, changing, and competitive process characterized by constant adsorption and desorption of proteins which involves hydrophobic interactions, ionic or electrostatic forces, and hydrogen bonds or Van der Waals forces [11,12].

Protein adsorption is dependent on various physiochemical characteristics of the blood-contacting surface including the intrinsic properties of the material as well as the concentration, relative affinity, and diffusion coefficient of plasma proteins [13]. It has been envisioned as a two-step process. First, large and more abundant plasma proteins (albumin, fibronectin, fibrinogen, factor XII, high-molecular-weight kininogen, and immunoglobulin G) that display a stronger attractive interaction than the smaller ones predominating in the plasma bulk [14] bind competitively and sequentially. There also occurs adsorption of low and medium molecular weight (MW) proteins in the body of the membrane, a slower and dynamic phenomenon depending on the membrane thickness and structure that is limited by membrane permselectivity and involves continuing enzymatic reactions and substrate replacement [6]. While protein adsorption onto a hydrophilic surface is usually weaker and more reversible, it is stronger and tends to be irreversible onto a hydrophobic surface [15]. Conformational changes may also occur once proteins are adsorbed onto biomaterial surfaces and acquire relevance for the ensuing biological responses that take place during the HD session [10].

The phenomenon of plasma protein adsorption markedly influences the performance characteristics of the HD membrane [16]. Besides governing the biocompatibility profile of the biomaterial, high adsorptive properties may contribute to clearance during the HD procedure of noxious compounds such as beta-2 microglobulin, complement factor D, and peptides [17]. Exactly how improving adsorption benefits blood depuration is not known [12]. Importantly, the adsorption phenomenon is highly nonselective while excessive protein adsorption can limit the diffusive and convective capacity of a membrane and hence reduce solute removal particularly with medium-large molecules [18].

Thus, it appears that any biomaterial for HD treatment should be carefully examined for its adsorptive properties, possibly by evaluating the amount, composition, and conformational change of the surface-adsorbed protein layer. One suitable approach for examining protein adsorption during HD is the application of proteomic techniques. Several studies have shown that proteomics allow one to investigate in an unbiased manner the full adsorption potential of dialysis membranes [12,19,20,21]. Current proteomic approaches not only enable one to quantify and identify all proteins present in the eluate obtained by the hemodialyzer but also to characterize the metabolic pathways and biologic networks affected by the proteins detected using bioinformatic tools, unraveling their pathophysiological significance [22]. Proteomics is seen as a possible tool for formulating new hypotheses on markers of interest in HD [12].

### 2.2. Activation of Blood Cascades and Cells

All blood-contacting medical devices can trigger either coagulation, resulting in clot formation, or immune reactions [23]. HD membranes are inherently pro-coagulant and promote coagulation through the activation of several inter-connected processes following protein adsorption; these include the generation of thrombin, the adhesion and activation of platelets and leukocytes, and the activation of complement (Figure 1) [23]. 

A brief description of these processes is provided in the next sections. The problem of biomaterial-associated thrombogenicity has been fully discussed in a series of four recent reviews [10,23,24,25].

#### 2.2.1. Coagulation Cascade

Adsorption of contact system components onto HD membrane material facilitates activation of the intrinsic coagulation pathway [26]. Autoactivation of adsorbed factor XII converts prekallikrein to kallikrein and starts coagulation and thrombin generation [23]. Thrombin promotes local platelet aggregation and acts on fibrinogen to form fibrin monomers, ultimately resulting in a platelet–fibrin thrombus which may foul the device and cause it to fail [23].

Given the tendency of blood–membrane interactions to activate a coagulation cascade, one resorts to anticoagulation in order to avoid the risk of premature clotting or premature termination of the HD session. Systemic anticoagulation with unfractionated heparin or low MW heparin is routinely employed and effective [27]. However, optimal anticoagulation remains a controversial issue for clinical practice [28], and use of anticoagulants may increase the uremic bleeding tendency due to platelet dysfunction. Thus, systemic anticoagulation is not indicated in patients at high risk of bleeding, including those suffering from acute bleeding, a recent head injury, and those scheduled for major surgery. Several alternative methods have been proposed, each with its own advantages and disadvantages [29,30], including the use of heparin-coated membrane as outlined in Section 3.2.1. There is still no universal agreement as to the precise anticoagulation for HD patients, particularly those needing special care. This is all the more reason for dialysis membranes that only minimally activate the coagulation cascade.

#### 2.2.2. Activation of Platelets

Platelet activation also causes thrombogenicity of HD membranes following blood–membrane interaction. Platelet activation occurs through adhesion to proteins (mainly fibrinogen) adsorbed onto the membrane and indirectly via biomaterial-induced activation of the coagulation system and others [25]. Platelets bind to surface-adsorbed fibrinogen via the integrin receptor GP IIb/IIIa. Conformational changes of adsorbed fibrinogen resulting in the exposure of platelet binding regions is relevant to this process [10]. Adherent platelets become activated and begin to spread and lose their shape. Simultaneously, they expose on the outer surface procoagulant phospholipids like phosphatidylserine and phosphatidylcholine which bind to plasma coagulation factors [8]. This procoagulant activity and the secretion of granular content leads to the aggregation of platelets such as to form the platelet–fibrin mesh of the clot or thrombus [8].

Surface modifications designed to suppress platelet activation and subsequent biological responses are a current trend in membrane development. 

#### 2.2.3. Complement Activation

Activation of the complement system in HD has long been known [31]. Mechanisms triggering complement activation may be direct or indirect and involve the classic alternative and lectin pathways [32]. Complement initiators can directly bind to the biomaterial. The indirect mechanisms proposed include binding of immunoglobulin G initiating the classic pathway, activation of the lectin pathway by carbohydrate structures or acetylated compounds, or activation of the alternative pathway by plasma protein-coated biomaterials [32].

Activation by biomaterial of the complement cascade leads to C3 cleavage, forming C3a and C3b. Increased levels of this last generate C5-convertase which cleaves C5 into C5a (a powerful anaphylatoxin like C3a) and C5b. Finally, the binding of C5b to the surface and interaction by it with C6-C9 results in formation of the membrane attack complex (MAC/C5b-9) [33]. Note that biomaterials not only activate the complement cascade but may also affect its course, mainly by binding complement inhibitors such as Factor H or C1q inhibitor to the biomaterial surface [34].

Activation of the complement system leads to the generation of effector molecules triggering several biological responses [33]. Short-term effects include inflammation and promotion of coagulation [32]. Generation of anaphylatoxins C3a and C5a causes leukocyte recruitment and activation, with the ensuing release of reactive oxygen species (ROS; oxidative burst), pro-inflammatory cytokines and chemokines, and granule enzymes such as elastase and myeloperoxidase [32]. In addition, activated complement-induced upregulation of complement receptor 3 (CR3) on leukocytes causes them to bind to C3 fragments (iC3b) deposited onto the membrane, leading to leukopenia [8]. CR3 on neutrophils is also important for the formation of circulating platelet–neutrophil aggregates that are pathophysiologically relevant in both inflammatory and thrombotic processes [35]. Furthermore, generation of C5a during HD leads to tissue factor and granulocyte colony-stimulating factor being expressed in neutrophils, shifting to a procoagulant state [36], while on the other hand, complement activation is impacted by the coagulation system [37].

Long-term complications of HD such as cardiovascular events and infection have been linked to complement activation. Particularly, by causing inflammation and coagulation, complement may contribute to HD patients’ susceptibility to cardiovascular disease [38]. Thus, intervention targeting the complement system may improve biocompatibility, dialysis efficacy, and patients’ long-term outcome [8,32].

#### 2.2.4. Activation of Leukocytes

During HD, myeloid leukocytes such as neutrophils and monocytes may adhere or be activated on a biomaterial surface. The main triggers for leukocyte activation are adsorbed proteins (fibrinogen, fibronectin, and iC3b being of primary importance) and adherent platelets [25]. Adsorbed and soluble complement products, as described in the previous section, represent the most prominent triggers of leukocyte activation. Kallikrein-induced activation also contributes, as well as adherence to already adsorbed and activated platelets eliciting outside-in signaling which brings about global neutrophil activation [39]. Products released from activated leukocytes lead to local tissue damage and to recruitment and activation of more inflammatory cells [25].

## 3. Effect on In Vivo Biocompatibility of Surface-Modified Membranes

Besides the actual blood composition, adsorption of plasma proteins onto dialysis membranes is related to the physicochemical properties of the polymer that constitutes the membrane [18,40]. The main characteristics influencing protein adsorption are listed in Table 1.

The evidence available indicates that each single dialysis membrane material has multiple and different characteristics that may contribute to interactions with blood components, including the adsorption of proteins [20]. To improve the biocompatibility profile of hemodialyzers, it is essential that the surface material be modified so as to attenuate clotting as well as innate immune system activation by altering the blood–material interface [23]. Several surface modification approaches have attempted to modulate blood–material interactions using physicochemical methods or surface biofunctionalization [8,24,41,42,43].

In the following sections, we report the effects of recently developed surface-modified membranes on biocompatibility during clinical HD.

### 3.1. HD Membranes Surface-Modified via Physicochemical Approaches

Physicochemical methods affect protein–cell–material interactions by modulation of surface charge, topography, and hydrophobic or hydrophilic interactions [23]. Several novel surface-modified membranes have been fabricated through intervention on the physicochemical properties of parent polymers with a view of mitigating the effects of blood incompatibility in HD.

#### 3.1.1. Asymmetric Triacetate Membrane

Cellulose triacetate (CTA) is a polymer characterized by replacement of the hydroxyl group of cellulose with an acetate group. CTA membrane has high diffusion efficiency and better antithrombotic properties than other synthetic membranes, which is related to its lower protein adsorption capacity [44]. To improve the performance characteristics of CTA membrane, a new asymmetric triacetate (ATA) membrane was developed by modifying CTA material. Asymmetric structure refers to the pore size distribution (smaller-sized pores on the internal surface and large pores on the external support layer) which is different from the homogeneous structure of parent CTA polymer. In addition, ATA membrane has a smoother surface than that of CTA, as shown by atomic force microscopy [44].

This last characteristic of ATA membrane (Solacea dialyzer; Nipro, Osaka, Japan) is expected to affect the protein adsorption process during HD. Preliminary studies in vitro have demonstrated lower protein adsorption onto ATA membrane than onto CTA membrane [44] or a polysulfone membrane with the same inhomogeneous pore size distribution [45]. In a prospective cross-over study carried out in four chronic HD patients, we used a bottom-up shotgun proteomics approach to examine the dialytic performance of ATA membrane as compared to the conventional parent symmetric polymer [46]. We used proteomic analysis to identify both the proteins adsorbed onto the hemodialyzer and the proteins recovered in the ultrafiltrate during the HD session. ATA membrane displayed a lower protein adsorption rate with a lower mass distribution pattern in the proteinaceous material. Moreover, the average number of proteins identified in the ATA eluate was significantly lower than from CTA, thereby confirming the overall tendency of adsorbed proteins to concentrate, primarily related to the smoother blood-contracting surface of ATA membrane [46]. However, the protein repertoire in the ultrafiltrate obtained during HD proved to be quite similar. Use of bioinformatics tools (ingenuity pathway analysis) to characterize the most relevant and highly represented metabolic and signaling pathways associated with proteins identified in the eluates obtained from the hemodialyzer at the end of the HD session showed that ATA retained fewer proteins which have a role in the canonical pathway of the coagulation system, prothrombin activation pathways, and the complement system (C3 complement factor). These results indicated that the newly developed ATA membrane has an improved biocompatibility profile vis-à-vis parent CTA membrane, the latter already being characterized [20] by its good biocompatibility.

The ATA membrane’s low tendency to activate the coagulation cascade was confirmed in subsequent clinical investigations. Vanommeslaeghe et al. [47] compared seven different settings in a cross-over study in ten HD patients at their midweek session: ATA membrane with regular and half doses of anticoagulation; a conventional polysulfone membrane with regular, half dose, and half dose plus priming of the dialysis circuit with a human albumin solution; and heparin-coated polyacrylonitrile membrane without systemic anticoagulation or anticoagulation plus priming with human albumin. The outcome parameter was the dialyzer fiber patency at the end of the HD session, visualized by a three-dimensional micro-computer tomography technique which allows one to assess fiber blocking at the single fiber level [48]. This is important to know since reduction of the whole exchange surface of a dialyzer by fiber clotting impairs dialysis efficiency. The results indicated that in fiber patency, ATA membrane was superior to all the other dialysis procedures and that the open fibers post-HD were not affected by reducing anticoagulation to half the regular dose with ATA [47].

That the ATA membrane outperformed a high-flux polysulfone membrane in avoiding clotting (as expressed by the relative number of open fibers at the end of a post-dilution hemodiafiltration session) was confirmed in a randomized cross-over study where patients received either a regular or half dose of anticoagulation [49]. Two further clinical studies showed the possibility of a safe and successful further reduction of anticoagulation during dialysis [50,51]. In a cross-over study, ten patients on chronic HD underwent their midweek session in three different dialysis modes (pre-dilution hemodiafiltration, HD, and post-dilution hemodiafiltration) using one-quarter or zero anticoagulation [50]. When one-quarter anticoagulation was applied, the number of open fibers post-dialysis proved almost optimal whatever the dialysis modality. With zero anticoagulation, fiber blocking was more prominent but on the limited side, with no premature termination of the dialysis session [50]. Vandenbosch et al. examined in a phase II pilot two-arm cross-over study whether it was possible to abolish systemic anticoagulation completely when using ATA membrane [51]. In arm A, patients were dialyzed with ATA dialyzer in combination with a citrate-containing dialysate while in arm B, ATA was on the high-volume predilution hemodiafiltration modality. The success rate in completing the extracorporeal procedure without preterm clotting was 90.8/94% in arm A and 83.3/86.2% in arm B (intention to treat/as treated, respectively).

Overall, the evidence available indicates that the ATA membrane has excellent biocompatibility in terms of its low tendency to activate the coagulation cascade. Use of ATA appears to be a promising, safe, and efficacious strategy in patient conditions which need reduced or even zero systemic anticoagulation [47,49].

#### 3.1.2. Polymethylmethacrylate NF Membrane

Polymethylmethacrylate (PMMA)-based HD membranes have been developed by controlling the stereocomplex structures that are formed from a mixture of syndiotactic PMMA and isotactic PMMA polymers. PMMA membrane has a homogeneous pore structure that provides a high specific surface area and a high protein adsorption property allowing removal of high MW proteins that are not efficiently removed by HD or hemodiafiltration [52]. However, proteins adsorbed onto PMMA membrane undergo structural changes that cause platelet adhesion and activation [53]. To inhibit this last while maintaining adsorption performance, a new PMMA membrane dialyzer, NF, was developed. 

One major factor in the structural changes to adsorbed protein consists in the structure of the water adsorbed onto the membrane surface (adsorbed water). Since changes in the structure of adsorbed water may disturb the hydration structure of proteins adsorbed (which preserves their structure and activity), it is important to counteract any factors that trouble the adsorbed water [53]. Compared to conventional PMMA, the new PMMA NF membrane features a reduction in the negative charge, a characteristic designed to decrease the electrostatic interactions from the membrane and prevent disturbance of the adsorbed water. Use of attenuated total reflection Fourier transform infrared spectroscopy (ATR-FTIR) demonstrated that the structure of adsorbed water on the NF membrane surface was close to that of free water and the structure of proteins adsorbed was close to their native state [53]. Modification of the surface structure of PMMA NF resulted in vitro, as compared to conventional PMMA, in a 100-fold decrease in platelet adhesion and a remarkable reduction of fibrinogen adsorption and leukocyte activation while maintaining almost equal adsorption properties [53].

Two clinical investigations have been performed with the PMMA NF membrane (Filtryzer NF dialyzer; Toray Industries, Inc., Tokyo, Japan). Masakane et al. compared in vitro and in vivo the new PMMA NF membrane to conventional (BG series) PMMA [54]. Studies in vitro confirmed the remarkable lower platelet adhesion and activation with PMMA NF, yielding a nearly equivalent MW of adsorbed proteins between the two materials as assessed by electrophoresis. In a cross-over in vivo study, six chronic HD patients were treated for three consecutive months with either PMMA NF or conventional PMMA. With the latter membrane, unlike PMMA NF, a decrease in platelet count during the dialysis session was found [53]. Activation of platelets and their following tendency to aggregate with other platelet cells and leukocytes might also explain the significant decrease in skin perfusion pressure of the sole of the right foot (as a measure of peripheral circulation) detected during HD with the conventional PMMA, which in turn might cause some dialysis-related side effects [54]. Peripheral circulation proved to be more stable without significant changes when patients were dialyzed with the PMMA NF membrane [54]. Uchiumi et al. studied thirty-seven patients on maintenance HD who were randomly allocated to dialysis using PMMA NF (*n* = 18) or a polysulfone membrane (*n* = 19) for one year [55]. While the primary end-point of the study, C-reactive protein values, did not change throughout the study period, the platelet count significantly decreased after 6 and 9 months with the polysulfone membrane, and it was stable with PMMA NF membrane. Moreover, at the same time points, dialysis with polysulfone was associated with a significant decrease in the creatinine generation rate (% CGR), an index for estimating muscle mass in dialysis patients [56]. By contrast, % CGR was stable with PMMA NF membrane which might indicate the usefulness of this surface-modified PMMA in maintaining the patient’s nutritional condition, though this issue requires further evidence [55].

#### 3.1.3. Hydrophilic Polysulfone Membrane

Polysufone is one of the most widely used materials for HD membranes due to its properties including excellent performance in solute removal, thermodynamic stability, chemical inertness, and mechanical strength. However, use of polysulfone is restricted by poor hydrophilicity and blood compatibility [57].

To improve the biocompatibility of polysulfone material, a new polysulfone-modified membrane was developed though application of a new hydrophilic polymer (Hydrolinik^TM^ NV) onto the inner surface of the polysulfone material [58]. The modification was designed to enhance the mobility of water adjacent to the membrane surface so as to reform the membrane surface. Water plays a central role in mediating the adsorption of proteins and other biomolecules onto the surface of materials [10,13]. Activation of the coagulation cascade and complement system can be minimized on charge-neutral strongly water-binding surfaces with a high substrate surface mobility [10].

The new NV membrane (Toraylight NV dialyzer; Toray) was specifically designed aiming at antifouling and antithrombogenic effects during HD [58]. Yamaka et al. reported lower platelet activation and adhesion to the membrane surface when six patients on maintenance HD were dialyzed for two weeks with NV membrane as compared to the following two weeks using a conventional polysulfone membrane [59]. Hidaka et al. showed a reduction in levels of platelet-derived microparticles (which are significantly increased in prothrombotic diseases) upon use of NV for 3 months [60]. Koga et al. examined the activation of blood cells by five different brands of polysulfone membrane in an in vitro system using mini module dialyzers [61]. Use of NV dialyzer was associated with lower adhesion and activation of platelets and slight activation and production of ROS by neutrophils. This improved biocompatibility of NV membrane was related to the observed lower adsorption of fibrinogen onto its surface which mediated GP IIb/IIIa platelet activation and Mac-1/alphavbeta3 neutrophil activation [61]. In a similar in vitro study model, the same group reported platelet activation, formation of platelet–neutrophil complexes, and neutrophil ROS production with a conventional polysulfone membrane, whereas NV membrane induced slight or no cell activation [62]. Ronco et al. in a prospective randomized clinical study examined the antithrombogenic effects of NV dialyzers compared to a conventional polysulfone, each membrane being used for 6 months [63]. After 3 weeks of treatment, the authors conducted a heparin reduction test for 5 weeks to evaluate the minimum amount of anticoagulation needed to perform the HD session safely. More patients in NV group reached a marked reduction in heparin dosage without clotting events than was the case in the control group. The tendency toward less clotting with the NV membrane kept with the cumulative clotting score adjusted for the heparin percentage used [63].

Other studies examined the effect of the NV membrane-containing dialyzer on some complications frequently encountered in patients on maintenance HD.

Hypotension during the HD session is a common side effect (intradialytic hypotension; IDH). It is caused by a failure to compensate reduced circulating blood volume and is associated with increased mortality. It is also defined by a fall in systolic blood pressure (SBP) > 20 mmHg from baseline or by the appearance of symptoms along with hypotension calling for medical intervention [64]. Tsuchida et al. examined whether the new NV membrane provides any advantage over conventional polysulfone on IDH [65]. In a prospective stratified-randomized multicenter trial, forty diabetic patients on chronic HD were treated for six months with either NV or conventional polysulfone or polyethersulfone dialyzers. The use of NV dialyzers significantly increased post-HD SBP and the lowest SBP during the HD procedure compared to the control group. It also required less intervention for IDH, while factors potentially influencing IDH (pre-dialysis body weight and ultrafiltration rate during HD) proved similar throughout the study period. The mechanisms behind the NV effect on the hemodynamic status remain to be clarified. The authors hypothesized that the improved biocompatibility of the NV membrane with its suppressed activation of platelets and leukocytes might relieve inflammation and oxidative stress during HD, with an ensuing improvement in endothelial function and recovery of the vasoconstriction response to hypotension [65]. Improvement of endothelial function assessed by flow-mediated dilation was previously reported after use of NV dialyzer for 3 months [60]. 

Anemia is a common complication in HD patients and is associated with many adverse clinical consequences [66]. Administration of erythropoiesis-stimulating agents (ESAs) represents the mainstay treatment. Not all HD patients, however, have a good response to ESA, a condition called ESA resistance requiring higher doses of ESA that may be associated with unfavorable outcomes including increased cardiovascular risk [67]. Hyporesponsiveness to ESAs is primarily due to the systemic inflammation of CKD, known as inflammation anemia [68,69]. Activation of immune system mediators induces inflammation anemia by impairing erythropoiesis, reducing erythrocyte survival, and restricting iron absorption [69]. A key role is played by Interleukin-6 (IL-6), a pro-inflammatory cytokine that induces the expression of hepcidin, an iron regulatory hormone [68]. The role of IL-6 was well evidenced by a recent randomized placebo-controlled phase one and two trial in inflamed HD patients with elevated IL-6 levels [70]. Use for 12 weeks of a novel anti-IL6 ligand antibody, ziltivekimab, reduced the ESA requirement and resistance index (ERI; calculated as U/kg per g/dL hemoglobin), decreased markers of inflammation, and increased serum albumin [70]. 

Management of inflammation anemia in patients suffering from renal failure remains quite challenging [71]. Kakuta et al. randomized twenty HD patients with high IL-6 concentration to NV or conventional polysulfone dialyzers, assessing IL-6 removal performance during dialysis and the one year effect on erythropoiesis and nutritional status [72]. Both dialyzer types were associated with a significant dialysis-associated production of IL-6, which proved to be lower with the NV dialyzer as indicated by the higher removal rate and the lower concentration ratio at the end of the HD session. No differences between the two groups of patients were found in either ESA dosage or ERI. However, the overall changes of these two parameters from baseline were increases with conventional polysulfone and decreases with NV dialyzer, suggesting that NV might reduce the risk of ESA hyporesponsiveness [72]. A subsequent retrospective analysis [73] examined the one year ESA requirement in HD patients treated with NV (*n* = 122) or conventional polysulfone (*n* = 129). While both ESA dose and ERI were unchanged with the NV dialyzer, ERI increased with conventional polysulfone. In addition, a significant decrease in ESA dose and ERI was found in highly ESA-resistant patients of the NV group [73]. The different results between the two membranes may be related to the reduced dialysis-associated acute IL-6 induction by NV [72]. The authors conjectured that dialysis-associated acute IL-6 induction induced a transitory increase in IL-6 levels, affecting the erythropoiesis system at each HD session and aggravating ESA resistance over time [73]. While the potential benefits on NV membrane’s ESA resistance is an intriguing finding, limitations of both above studies [72,73] hamper any firm conclusion from being drawn and call for further investigation.

#### 3.1.4. Surface-Modifying Macromolecule—Modified Membrane

A more recent approach to reducing blood-contacting surface interactions and hence the need for anticoagulation during dialysis uses a novel polysulfone membrane obtained by mixing basic polymeric material with surface-modifying molecules [74,75]. Surface-modifying molecules (SMMs) such as the Endexo^TM^ (Interface Biologics, Inc., Toronto, ON, Canada) family are fluoro-oligomeric additives that are able to migrate to the surface, thus providing passive surface modification which suppresses adsorption-induced conformational changes to procoagulant proteins and reduces platelet adhesion and activation [76]. Currently, Endexo additives are approved for use in peripherally inserted central venous catheters [76,77]. A new SMM-modified membrane for use in HD (Optiflux Enexa dialyzer; Fresenius Medical Care, Waltham, MA, USA) has been developed which incorporates SMM1, a modifying molecule belonging to the Endexo family, into the dialyzer fibers, mixing it with the basic polymers of a standard polysulfone membrane material (Optiflux Advanced Fresenius Polysulfone) [74]. Surface characterization of the inner lumen showed that the SMM1-modified membrane is more hydrophobic than standard polysulfone (whilst maintaining membrane structure and surface hydrophilicity) and has a reduced absolute surface charge. In vitro biocompatibility of SMMA-modified membrane proved better than standard polysulfone, as measured by significantly reduced platelet adhesion and activation and clotting activation, potentially reducing thrombus formation [74].

The first clinical experience of the novel dialyzer containing an SMM1 fluorinated polyurethane was a prospective, sequential, and open-labeled study [75], carried out to meet the US Food and Drug Administration guidance for clinical performance testing of a new dialyzer [78]. After completing 12 dialysis sessions with a standard polysulfone dialyzer, HD patients received 38 sessions with the Endexo-modified membrane dialyzer (*n* = 664 treatments over 13 weeks). There was good tolerance to the device and no related serious adverse events. No overt evidence of complement activation (C3a, C5a, and sC5b-9) was found during dialysis with either dialyzer. Performance of the SMMA-modified dialyzer proved generally similar to that of unmodified standard polysulfone, though the removal rate of beta2-microglobulin achieved with the modified dialyzer was higher (i.e., 68% with correction) [75]. Interestingly, lower heparin requirements and reduced ESA doses were observed during the study period with the modified dialyzer. However, the lack of protocol-driven modifications for HD treatment-related modifications does not allow any definitive conclusion. Additional longer-term clinical studies are required to provide insight into the clinical implications of SMMA-modified dialyzer [75].

### 3.2. HD Membranes Surface-Modified by Biofunctionalization

Surface biofunctionalization seeks to endow the surface with the capacity to interact actively with cell surface receptors or enzymatic activation [79]. Surface modification by biofunctionalization includes coating of surfaces with biofunctional entities which can be applied passively or covalently linked to the surface (23). However, covalent attachment of pharmaceutical agents like anticoagulants during the production of a membrane or dialyzer is inconceivable [8]. Passivating modifications to the material surface attempt to minimize the interaction with blood defense systems [24]. The coated dialyzer membranes used in HD are described below.

#### 3.2.1. Heparin-Coated Membranes

Heparin-coated membranes were realized for patients at high risk of bleeding. One example is heparin coating of AN69 ST membrane, a polyacrylonitrile sodium methallylsulfonate copolymer coated with high-MW polyethyleneimine before heparin grafting to reduce electronegativity of the original AN69 polymer, resulting in the capacity to bind heparin to its surface [6].

Observational and uncontrolled studies showed promising results with heparin-coated membranes, albeit less confirmed by controlled studies. In experimental in vitro and in vivo models, AN69 ST membrane coated with heparin demonstrated a sustained anticoagulation property [80]. Use of AN69 ST membrane allowed a 50% reduction of the heparin dose with no increased risk of massive clotting [81]. Lavaud et al. reported successful HD sessions without systemic administration of heparin in either stable or bleeding-risk HD patients [82].

A precoated heparin-grafted membrane differing from the previous design was then manufactured [80,81,82] since it does not require any separate preparation step and allows priming of the extracorporeal circuit identical with conventional membranes: the HeprAN membrane (heparin-grafted AN69 ST; Evodial dialyzer, Baxter, Deerfield, IL, USA).

The HepZero study was a controlled randomized study monitoring up to three heparin-free dialysis sessions in 251 maintenance HD patients at increased risk of hemorrhage, randomly allocated to a heparin-coated membrane (Evodial) or to “standard-of-care” heparin-free dialysis, defined as regular saline flushes or predilution hemodiafiltration [83]. The study’s primary end-point, successful completion of the first dialysis session, was achieved in 68.5% of patients randomized to the heparin-grafted membrane and in 50.4% of patients treated with standard of care. Use of heparin-coated membrane proved not to be inferior to saline infusion, though superiority could not be demonstrated. Furthermore, clotting events and dialysis efficiency were not reported [83]. Similar efficacy with heparin-grafted membranes was found in a French study [84]. In a multicenter randomized cross-over study in thirty-two long-term HD patients, Islam et al. compared heparin-coated to vitamin E-coated membrane dialyzers as a 4 h heparin-free HD strategy [85]. The primary end-point, represented by the percentage of HD sessions (*n* = 4 with each dialyzer: two with reduced heparin dose and two without anticoagulation) without a clotting event causing premature interruption of the session, was 78% with vitamin E-coated membrane and 81% with heparin-coated membrane. Both dialyzers exposed patients to a failure rate judged unacceptable by the authors [85].

More recently, a randomized cross-over study was carried out in six dialysis patients with thrombocytopenia, each dialyzed at the midweek session without anticoagulation using either a heparin-coated (Evodial dialyzer) or a polysulfone membrane [86]. Dialyzer performance was evaluated by assessing the number of open fibers at the end of the HD session using the micro-CT scan technique. The heparin-coated dialyzer resulted in substantially fewer coagulated fibers and a better subjective visual assessment score of fiber clotting. Moreover, no leaching of heparin resulting in undesired systemic anticoagulation was found [86]. While this study showed that heparin-coated membrane is superior to a non-coated membrane (polysulfone) in a 4 h HD session with no systemic coagulation, others have shown that fiber patency is not different from a polysulfone membrane using half the regular anticoagulation dose or is even inferior to other dialyzers for heparin-free HD [47].

Overall, the rate of clotting in anticoagulant-free HD using heparin-coated dialyzer is considered to be too high [85]. Use of membrane coated with heparin may be superior to saline flushes alone [83,84] but seems less effective than other heparin-sparing modalities [47,87]. To be sustainable, routine use of heparin-coated membrane dialyzer would probably require either circuit priming with an anticoagulant or systemic anticoagulation [47,81,85,88].

A combined anticoagulation strategy might help to decrease the rate of clotting observed with heparin-coated membrane. This is suggested by the results of the CiTED (Citrate and EvoDial) study [89]. In the randomized cross-over study, HD patients (*n* = 25) were treated with a combination of heparin-grafted membrane (Evodial) plus citrate-containing dialysate or regional citrate anticoagulation (RCA). Citrate-containing dialysate, besides the advantage of being an acetate-free dialysate, can reduce (though without abolishing) the need for heparinization during HD [89]. In turn, RCA is a well-studied and efficacious approach to reduce heparin exposure but is laborious, may cause metabolic disturbances, and is associated with higher costs than conventional HD. It was previously shown that RCA is superior to heparin-coated membrane in HD patients at risk of bleeding [87]. By contrast, in the CiTED study, the primary outcome (preterm interruption of prescribed HD sessions due to clotting) comprised 5.7% of sessions in the CiTED arm and 6.2% sessions in the RCA group, thereby meeting the non-inferiority criterion. Thus, combining a heparin-grafted membrane with a citrate-containing dialysate may be a valid and easy-to-use alternative to RCA in dialysis patients at high risk of bleeding [89].

Finally, heparin-coated membrane might unexpectedly reduce the concentration of pro-inflammatory cytokines [90]. Nineteen stable HD patients were first dialyzed with conventional membranes and enoxaparin as anticoagulant then with heparin-coated membrane without systemic anticoagulation. After the HD session with Evodial dialyzer, plasma levels of monocyte chemoattractant protein 1, endostatin, and activin A were 2–3-fold lower than with standard dialysis. Between-anticoagulant differences proved significant over time for all three cytokines [90]. By reducing the level of pro-inflammatory cytokines, heparin-free dialysis with heparin-coated membrane might improve the endothelial function, a hypothesis that needs to be explored.

#### 3.2.2. Vitamin E-Coated Membranes

ESRD patients on HD have increased levels of oxidative stress resulting from the reduced antioxidant defense given their uremic status and intradialytic generation of ROS [91]. Oxidative stress contributes to atherosclerosis via oxidation of low-density lipoproteins [92]. It is now thought to be a crucial hallmark and early causative factor of CV disease [92,93].

Oxidative stress in HD patients has been associated with CV disease progression [94]. Moreover, a reciprocal activation between oxidative stress and inflammation may occur [95]. The relevance of oxidative stress and chronic inflammation as contributors to atherosclerosis and CV mortality in HD patients is increasingly apparent [96]. Thus, antioxidant therapy may offer a promising strategy to reduce CV risk in patients on maintenance HD [97].

Vitamin E supplementation may provide antioxidant and anti-inflammatory effects in patients on HD. Vitamin E is a lipid-soluble antioxidant composed of eight compounds with high anti-inflammatory properties [98]. Two recent systematic reviews and meta-analyses examined the effects of vitamin E oral supplementation in HD patients [95,99]. Significant decreases in biomarkers of oxidative stress (malondialdehyde) [95,99], vascular inflammation (ICAM-1 and VCAM-1), and systemic inflammation (C-reactive protein) [95] were found. Reservations over these findings have been voiced, however, due to the high levels of heterogeneity observed between studies. A more effective strategy than oral administration to provide antioxidant defense by vitamin E is represented by vitamin E-coated membranes [100].

Vitamin E-modified membranes were designed to reduce oxidative stress in HD patients and improve biocompatibility [5]. The first vitamin E-coated membrane was cellulose-based (Excebrane dialyzer; Terumo Corporation, Tokyo, Japan). The results of in vitro and in vivo studies showed much better biocompatibility than the original regenerated cellulose membrane [101]. Later, to achieve the synergistic effect of the biocompatibility of synthetic membranes and the antioxidant activity of vitamin E, polysulfone-based vitamin E-coated membranes were developed and introduced in the market (initially by Terumo Corporation; subsequently, using a new technique by Asahi Kasei Kuraray Medical, Tokyo, Japan with the membrane name of VitabranE). 

Bioactive vitamin E (alpha-tocopherol) present on the blood surface of the modified membrane acts as an ROS scavenger and seems to control oxidative stress and lipid peroxidation [102]. Many studies were carried out in HD to examine the effects of vitamin E-coated polysulfone membrane (ViE-m). Despite a wealth of data, however, there remained inconsistencies between clinical trials yielding conflicting results. To summarize the entire available evidence up to March 2016, D’Arrigo et al. performed a comprehensive systematic review and meta-analysis including sixty studies [103]. The main end-points of interest included biomarkers pertaining to oxidative stress, inflammation, and anemia status. 

The effects of ViE-m on oxidative stress were characterized by a significant improvement in sensitive clinical biomarkers of oxidative stress: plasma and erythrocyte malondialdehyde (MDA), thiobarbituric acid reactive substances, and plasma and blood erythrocyte vitamin E levels. These findings may hold clinical relevance since in HD patients, elevated MDA and other biomarkers of oxidative stress can contribute to morbidity and mortality [104], while lower MDA levels have been associated with reduced mortality in long-term evaluations [105]. The efficacy of ViE-m as an antioxidant was shown by a subsequent study examining the levels of genetic damage found in HD patients [106]. Oxidative stress has been postulated as a risk factor for the high levels of DNA damage reported in CKD patients [107]. Forty-six HD patients were randomized to either a vitamin E-coated membrane or conventional polysulfone for six months. At the end of the study period, a significant decrease in the levels of oxidative DNA damage was found in ViE-m treated patients. Vitamin E deficiency proved also to be corrected [106]. However, these results were not reproduced in a subsequent randomized study in HD patients lacking glutathione transferase M1 enzyme activity [108]. Such patients exhibited increased oxidative DNA damage and mortality compared to those with an active enzyme [109]. Use of ViE-m for three months did not show any benefit over standard polysulfone membrane on the primary study endpoint, represented by absolute changes in by-products of oxidative stress and inflammation [108].

As for inflammatory biomarkers, the meta-analysis indicated that use of ViE-m induced a significant decrease in circulating levels of IL-6 but not C-reactive protein [103].

A more recent study investigated the effect of ViE-modified membrane on hemodialysis inflammaging [110]. Inflammaging is a persistent, low-grade, sterile, and non-resolving inflammatory state which downregulates innate and adaptive immune responses in chronic disorders [111]. Sepe et al. conducted a randomized controlled cross-over study including eighteen ESRD patients, each treated for 3 months with low-flux HD with polysulfone membrane, low-flux HD with vitamin E-coated membrane, and pre-dilution hemodiafiltration with high-flux polysulfone dialyzer [110]. Inflammaging was assessed by changes in indoleamine 2,3-dioxygenase-1 activity (IDO1) and nitric oxide (NO) formation. Both parameters, involved in innate and adaptive immune response and in promoting chronic inflammatory disorders [112,113], were higher in HD patients than in healthy controls. Treatment with ViE-coated membrane resulted in a significant reduction of IDO1 activity and NO formation when compared to the other two extracorporeal procedures. Use of ViE-m could at least partially preserve against dysregulation of the immune system during HD [110].

Another major point of interest in the meta-analysis by D’Arrigo et al. pertains to the anemic status [103]. Use of ViE-m did not influence hemoglobin or hematocrit levels, erythrocyte count, or ESA dosage. However, it was associated with a significant reduction in ERI. This was confirmed in a controlled multicenter study randomizing 93 HD patients on stable ESA therapy to either ViE-coated polysulfone membrane or a low-flux synthetic dialyzer [114]. After adjusting for baseline ERI, mean ERI decreased in the ViE group and increased in the control group (*p* = 0.001). These results indicated that ViE dialyzer can improve ESA response, possibly though a decrease in the IL-6 level [114]. However, the impact of ViE-m on anemia control remains to be established since neither hemoglobin nor ESA dosage were affected [103,114].

A more recent clinical study [115] was carried out as part of a regulatory application [78]. Patients on maintenance HD were treated with conventional high-flux hemodialyzers for two weeks then switched to ViE-m for 36 HD sessions, thereafter returning to the conventional dialyzer. Removal of urea, creatinine, and beta 2-microglobulin, and ultrafiltration coefficients proved similar to conventional dialyzers. In terms of biocompatibility evaluation, no differences were observed regarding leukocyte or C3a fluctuations during the HD session. However, ViE-m showed a lower fluctuation in the platelet count. Interestingly, the frequency of IDH episodes was reduced during the phase of the study using ViE-coated membrane [115]. This finding is consistent with a previous study and might be related to the reduced oxidative stress induced by ViE-m [116].

In the aggregate, the available evidence indicates that coating with vitamin E improves the biocompatibility profile of membranes for HD, though the true potential of the modified membrane in the overall clinical management of HD patients remains to be elucidated.

A further improvement in the biocompatibility of vitamin E-modified membrane may be represented by enriching the material with alpha-lipoic acid (ALA) [117]. ALA is a fat-soluble antioxidant widely used as a drug in disorders associated with oxidative stress [118]. Following successful immobilization of ALA on polysulfone membrane surface [118], polysulfone-based membranes were coated with alpha-tocopherol and/or ALA via a spin coating technique (which provides a uniform distribution throughout the membrane) [117]. Better results in terms of biocompatibility parameters (including protein adsorption, complement activation, and platelet adhesion) and antioxidant capacity were associated with the use of membranes prepared with a combination of both vitamin E and ALA [117]. This new bioactive membrane now requires studying in vivo.

## 4. Conclusions

Hemodialysis is a life support therapy for a growing number of patients suffering from kidney failure. Survival and quality of life have improved compared with the past, but despite the considerable technical and scientific advances, results are still not satisfactory and high mortality and morbidity still characterize HD therapy [119]. The main determinant of the success and quality of HD is the artificial membrane contained in the hemodialyzer through its clearance and biocompatibility properties. Thus, there is an unmet need for better performing HD membranes in terms of more friendly biocompatibility and targeted selective removal to improve the dialysis outcome. 

The biocompatibility of the membrane material is undoubtedly a key factor for the therapeutic effect of HD, and modifications to dialysis membrane biomaterials should aim at ameliorating blood compatibility as a top priority [57]. Recognizing the importance of the physiochemical properties of the blood-contacting membrane surface, research has focused on making structural modifications or using bioactive compounds to mitigate the effects of adverse blood–membrane interactions that negatively impact the clinical performance. As reviewed in the present article, recent advances in surface modification of HD membrane materials already available on the market (Table 2) as well as under experimental development hold promise of an improvement in the biocompatibility profile, which may significantly affect on patient outcome. The role of these promising membranes in the overall management of patients on chronic HD, however, requires further clinical testing trials and longer follow-up. 

## Figures and Tables

**Figure 1 biomedicines-10-00844-f001:**
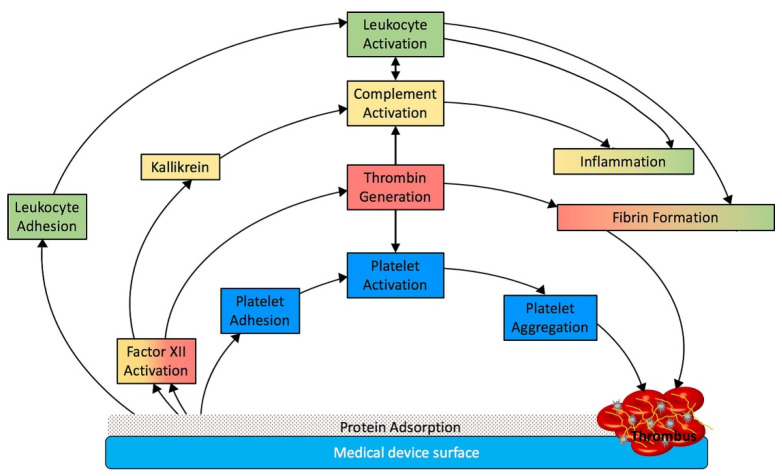
Activated pathways and their interplay during the interaction of blood components with artificial surfaces like the dialysis membrane material. Reproduced from reference [23] (with permission of authors and publishers).

**Table 1 biomedicines-10-00844-t001:** Physicochemical characteristics of hemodialysis membrane materials influencing protein adsorption.

Roughness
Micro domains and morphology
Charge
Crystallinity
Chemical composition
Hydrophilic/hydrophobic regions
Adsorbed water, proteins, and ions

**Table 2 biomedicines-10-00844-t002:** Summary of clinical studies on surface-modified membranes for hemodialysis.

Membrane	Surface Modification	Main Findings	Refs.
Asymmetric triacetate (ATA)	Smoother surface of parent polymer symmetric cellulose triacetate	Low protein adsorption Low tendency to activate the coagulation cascade with reduced anticoagulation during HD	Cross-over study [46] Cross-over studies [47,50] Randomized cross-over studies [49,51]
Polymethylmethacrylate NF	Reduction in negative charges of parent polymer polymethylmethacrylate	Stability of platelet count during HD	Cross-over study [54] Randomized study [55]
Hydrolink NV	Application of a hydrophilic polymer onto the inner surface of PSF material	Low platelet activation and adhesion to membrane Reduced platelet-derived microparticles Anti-thrombogenic effects Improved intradialytic hemodynamic status in diabetics May improve ESA resistance	Prospective sequential study [59] Randomized study [60] Randomized study [63] Stratified-randomized study [65] Randomized study [72] Retrospective study [73]
Surface modifying molecule-modified membrane	Incorporation of surface modifying molecule 1 into PSF dialyzer fibers	Safety; good removal of beta2-microglobulin	Prospective sequential study [75]
Heparin-coated membrane	Binding of heparin on the blood side of polyacrylonitrile sodium methallylsulfonate copolymer coated with polyethyleneimine before heparin grafting	Lower risk of bleeding but need for systemic anticoagulation not eliminated May reduce pro-inflammatory cytokines	Observational studies [81,82] Randomized controlled studies [83,84,87] Randomized cross-over studies [85,86] Randomized controlled study [90]
Vitamin E-coated membrane	Coating with vitamin E (alpha-tocopherol) on the blood surface of membrane	Decreased oxidative stress May have an anti-inflammatory effect Improvement of HD inflammaging May improve anemia Not inferior in anti-coagulation to heparin-coated membrane	Meta-analysis [103] Meta-analysis [103] Randomized controlled cross-over study [110] Meta-analysis [103] Randomized controlled study [114] Randomized cross-over study [85]

HD—hemodialysis; PSF—polysulfone; ESA—erythropoiesis stimulating agent.

## Data Availability

Not applicable.

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
