# Peer review of "Biocompatibility of Surface-Modified Membranes for Chronic Hemodialysis Therapy"

_biomedicines, 2022, doi:10.3390/biomedicines10040844_

Round 1
Reviewer 1 Report
In this manuscript, the authors reviewed the biocompatibility of surface-modified membranes for chronic hemodialysis therapy. Many different surface-modified membranes for hemodialysis have been manufactured over recent years by varying approaches in the attempt to minimize blood incompatibility. Their main characteristics and clinical results in hemodialysis patients are reviewed in the present article. In overall, this review manuscript is interesting but in order to consider publication, this work should be revised. The following comments should be addressed for the improvement of their manuscript.
Comment 1: The overall study aims for this review study about compatibility of surface-modified membranes for chronic hemodialysis therapy need to be further clarified in detail as compared to current conventional chronic hemodialysis therapy.
Comment 2: The various recent reports and their research findings on the “different type of surface-modified membranes fabricated through intervention on the physicochemical properties of parent polymers, with a view to mitigating the effects of blood-incompatibility in HD” should be summarized into a table form and discussed for better understanding.
Comment 3: The future direction/perspectives/challenges and commercialization aspects for this potential surface-modified membranes for chronic hemodialysis therapy can be further discussed in detail in the conclusion section.
Comment 4: The carefully English correction is necessary for the whole manuscript. Please check and revise accordingly.
Author Response
We thank the Reviewer for her/his positive evaluation of the manuscript. We are glad to submit a revised version that integrates all the comments raised during the review process.
Comment 1: The overall study aims for this review study about compatibility of surface-modified membranes for chronic hemodialysis therapy need to be further clarified in detail as compared to current conventional chronic hemodialysis therapy.
According to the suggestion of the Reviewer, we have added the following sentence at the end of the introduction section (page 2 lines 9-11): “The aim of the present article is to examine the biocompatibility performance of the new surface-modified membranes and their potential beneficial effects, as compared to conventional membranes, in patients on chronic HD therapy.”
Comment 2: The various recent reports and their research findings on the “different type of surface-modified membranes fabricated through intervention on the physicochemical properties of parent polymers, with a view to mitigating the effects of blood-incompatibility in HD” should be summarized into a table form and discussed for better understanding.
Also according to the comments of Reviewer 2, we have prepared a table (Table 2 in the revised manuscript) summarizing the review.
Comment 3: The future direction/perspectives/challenges and commercialization aspects for this potential surface-modified membranes for chronic hemodialysis therapy can be further discussed in detail in the conclusion section.
We have added the following sentence in the Conclusion section (page 14 lines 38-42): “Recognizing the importance of the physiochemical properties of the inner blood-contacting membrane surface, research has focused on making structural modifications or using bioactive compounds, to mitigate the effects of adverse blood-membrane interactions that negatively impact the clinical performance.” And we have modified the last sentence of the manuscript (page 14 lines 45-47): “The role of these promising membranes in the overall management of patients on chronic HD, however, requires further clinical testing trials and longer follow-up.”
Comment 4: The carefully English correction is necessary for the whole manuscript. Please check and revise accordingly.
English language of the whole manuscript has been revised.
Reviewer 2 Report
A very interesting and thorough review. I have only a few comments and minor suggestions.
p.5 l.42 – Instead of “with a carboxylic group.” it should be “with a carboxylate group” or “with acetic acid ester group” or “with acetate group”. Cellulose triacetate polymer does not have carboxyl (or carboxylic) groups. It has ester groups.
p.8 l.13 – “allowing removal of high MW proteins” – The authors should clearly define, somewhere at the beginning of the manuscript, what is being/should be removed from the blood during HD procedure, which are low MW toxic substances and water. Some studies indicate that the “profiles of various proteins in the blood of HD patients differ from those in normal subjects” and that removal of certain blood proteins (high MW substances) by adsorption on/in HD membrane may improve the therapy.
p.9 l.4 – “Polysulfone …excellent performance in solute removal, …” - What does this mean, exactly. If the authors think about flux/selectivity ratio, then it is mostly related to the porosity and pore structure, and less related to the material type.
p.9 l.5 – What is “thermodynamic stability”?
p.9. l.46 – “…polysulfone membrane modified by surface modifying molecule” – This sequence surely needs to be modified!
p.10 l.27 – “Biofunctional methods” – I have not heard about methods that are biofunctional. Compounds can be biofunctional, for example.
There is only one figure in the manuscript. It would be good to add one or two more, for visualization of the various methods of the HD membranes’ modifications. I would also recommend addition of a table summarizing the review.
Author Response
We wish to thank the Reviewer for her/his appreciation of our manuscript.
p.5 l.42 – Instead of “with a carboxylic group.” it should be “with a carboxylate group” or “with acetic acid ester group” or “with acetate group”. Cellulose triacetate polymer does not have carboxyl (or carboxylic) groups. It has ester groups.
We thank the Reviewer for the comment. “With a carboxylic group” has been replaced by “with an acetate group” (page 6 line 3).
p.8 l.13 – “allowing removal of high MW proteins” – The authors should clearly define, somewhere at the beginning of the manuscript, what is being/should be removed from the blood during HD procedure, which are low MW toxic substances and water. Some studies indicate that the “profiles of various proteins in the blood of HD patients differ from those in normal subjects” and that removal of certain blood proteins (high MW substances) by adsorption on/in HD membrane may improve the therapy.
In accordance with the Reviewer’s comment, we have modified the Introduction section as follows (page 1 Lines 36-40): “It allows removal by diffusion and convection of retained low molecular weight (MW) toxic substances and excess water - which accumulate in the blood because of kidney failure -, providing life support for patients to live with a variable degree of rehabilitation. Removal of high MW substances may also occur by adsorption on the surface of certain membranes.”
p.9 l.4 – “Polysulfone …excellent performance in solute removal, …” - What does this mean, exactly. If the authors think about flux/selectivity ratio, then it is mostly related to the porosity and pore structure, and less related to the material type.
Broadly, it means good solute removal by the polysulfone membrane.
p.9 l.5 – What is “thermodynamic stability”?
Thermodynamic stability refers to a system that is in its lowest energy state, or in chemical equilibrium with its environment. Thermodynamics influences protein adsorption behaviour.
p.9. l.46 – “…polysulfone membrane modified by surface modifying molecule” – This sequence surely needs to be modified!
The sentence has been modified as follows (page 10 lines 10.11): “..polysulfone membrane obtained by mixing basic polymeric material with surface modifying molecules”.
p.10 l.27 – “Biofunctional methods” – I have not heard about methods that are biofunctional. Compounds can be biofunctional, for example.
According to the comment of the Reviewer, we have removed “Biofunctional methods” replacing it with “Surface modification by biofunctionalization…” (page 10 line 43).
There is only one figure in the manuscript. It would be good to add one or two more, for visualization of the various methods of the HD membranes’ modifications. I would also recommend addition of a table summarizing the review.
We have added a table (Table 2) which summarizes our review.
Reviewer 3 Report
Authors present a very comprehensive review about the role of biocompatibility of materials used for HD membranes production. Starting from molecular processes, Authors smoothly move into in vivo observations of novel, sometimes very new materials, which is a huge advantage of the manuscript.
Minor suggestions:
1) the article needs minor language edition ('dearth' on page 1 line 29; comma on page 4 line 19),
2) I suggest reducing the usage of some words, like 'uremic' on page 1 line 38;
3) when talking about coagulation cascade activation (page 3) you may already suggest heparin covering on the surface of some dialyzators to prevent that.
Author Response
We wish to thank the Reviewer for her/his appreciation of our manuscript.
Minor suggestions:
1) the article needs minor language edition ('dearth' on page 1 line 29; comma on page 4 line 19),
Dearth has been replaced by shortage (page 1 line 28); comma (page 4 line 16) has been removed.
2) I suggest reducing the usage of some words, like 'uremic' on page 1 line 38;
The word uremic has been removed (page 1 line 38).
3) when talking about coagulation cascade activation (page 3) you may already suggest heparin covering on the surface of some dialyzators to prevent that.
We thank the reviewer for the suggestion. We have added the following sentence (page 4 lines 5-7): “Several alternative methods have been proposed, each with its own advantages and disadvantages [29, 30], including the use of heparin-coated membrane as outlined in section 3.2.1.”
Round 2
Reviewer 1 Report
In overall, this manuscript was technically well revised. This revised manuscript meets the criteria of Biomedicines. Therefore, in my opinion, the revised manuscript can be accepted for publication.